# SynerGen: Contextualized Generative Recommender for Unified Search and Recommendation

## Abstract

The dominant retrieve-then-rank pipeline in large-scale recommender systems suffers from mis-calibration and engineering overhead due to its architectural split and differing optimization objectives. While recent generative sequence models have shown promise in unifying retrieval and ranking by auto-regressively generating ranked items, existing solutions typically address either personalized search or query-free recommendation, often exhibiting performance trade-offs when attempting to unify both. We introduce *SynerGen*, a novel generative recommender model that bridges this critical gap by providing a single generative backbone for both personalized search and recommendation, while simultaneously excelling at retrieval and ranking tasks. Trained on behavioral sequences, our decoder-only Transformer leverages joint optimization with InfoNCE for retrieval and a hybrid pointwise-pairwise loss for ranking, allowing semantic signals from search to improve recommendation and vice versa. We also propose a novel time-aware rotary positional embedding to effectively incorporate time information into the attention mechanism. *SynerGen* achieves significant improvements on widely adopted recommendation and search benchmarks compared to strong generative recommender and joint search and recommendation baselines. This work demonstrates the viability of a single generative foundation model for industrial-scale unified information access.

## 1 Introduction

Large-scale search and recommendation systems in e-commerce, short video, and food-delivery platforms are typically deployed as multi-stage cascades. A high-recall retriever (e.g., BM25 (Robertson et al., 2009), two-tower (Covington et al., 2016), or item–item collaborative filter (Linden et al., 2003)) narrows hundreds of millions of items to a few thousand candidates, which are then re-ordered by a compute-intensive ranker to optimize business metrics such as click-through rate (CTR), revenue, or watch-time. While pragmatic, this architecture forces each stage to optimize different losses, operate on disjoint features, and refresh on separate cadences—leading to redundant engineering and production misalignment.

Inspired by the success of large language models, recent work reframes recommendation as a generative problem: an autoregressive model directly outputs a ranked slate conditioned on user context, thereby aligning retrieval and ranking under a single objective. These generative recommenders have shown promising results in specific domains such as short-video feeds, food delivery, and even trillion-scale deployments.

However, existing systems typically target only one interaction mode. Query-aware search and query-free feed recommendation remain siloed: some models focus exclusively on search, while others excel in feed recommendation but ignore queries. Attempts at unification reveal an inherent trade-off: semantic query signals and collaborative behavioral signals compete within shared representations, so improving one task often degrades the other. To date, no published work demonstrates a single generative backbone that simultaneously excels at both retrieval and ranking *and* supports both query-aware search and query-free recommendation.

We address this gap with *SynerGen*, the first decoder-only generative recommender that unifies these four capabilities. Retrieval and ranking are jointly optimized within one backbone: retrieval with an InfoNCE loss (Oord et al., 2018) (using in-batch and mined hard negatives), and ranking with a hybrid pointwise–pairwise loss. Semantic, behavioral, and temporal signals are fused before the first Transformer layer, ensuring richer contextualization than late concatenation. Joint training allows semantic cues from queries to strengthen feed recommendation and collaborative signals to benefit search.

By demonstrating that a single generative foundation model can deliver state-of-the-art performance across both retrieval and ranking, in both search and recommendation, while meeting strict latency constraints, *SynerGen* positions generative recommendation as a universal backbone for industrial information access.

**Contributions.** In summary, we make the following contributions: 1. We present the first decoder-only generative model that seamlessly supports both query-aware search and query-free recommendation, achieving superior results on standard benchmarks. 2. We show that retrieval and ranking can be jointly optimized within a single backbone without cross-stage misalignment, simplifying deployment while improving effectiveness.

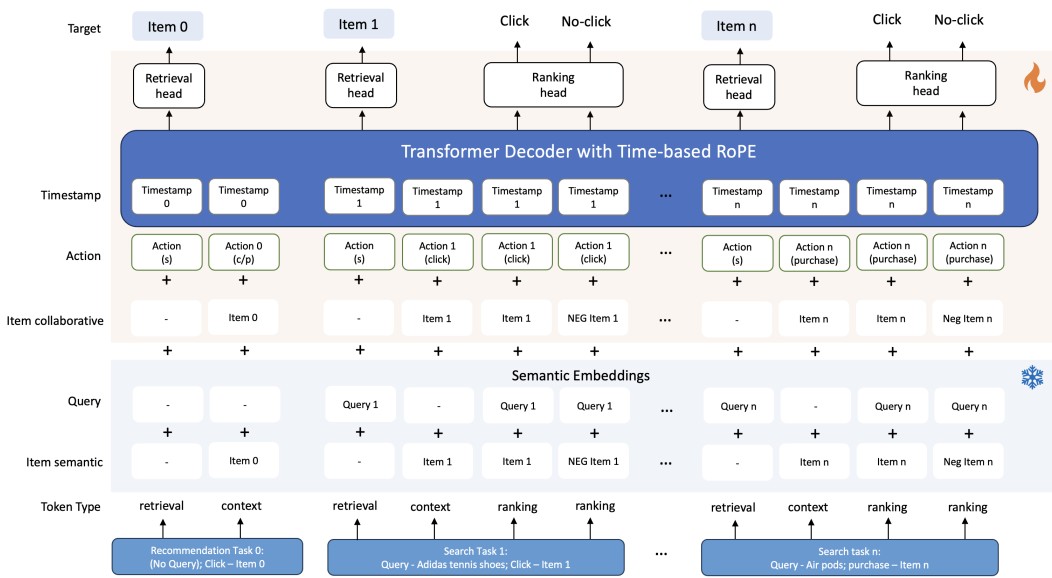

Figure 1: Overview of *SynerGen* pretraining, illustrating unified retrieval and ranking tasks within a joint search and recommendation framework. A single forward pass processes a user behavior sequence of length $n$, with context and task tokens jointly encoded. Semantic embeddings cached from a pretrained language model are frozen during training.

## 2 BACKGROUND AND RELATED WORK

As large generative models emerge in recommendation and search, the focus of applications has shifted toward addressing long-standing challenges such as: unifying search and recommendation, modeling evolving user preferences, and incorporating temporal patterns into sequential prediction.

**Unifying Search and Recommendation.** Traditional search and recommendation systems have been developed in silos—search engines retrieve items based on explicit queries, while recommenders predict user preferences from interaction histories. Such separation ignores the shared users and overlapping item spaces across tasks. Recent work has sought to unify search and recommendation into a single modeling framework (Zhao et al., 2025; Liu et al., 2024), enabling richer user representations, improved generalization, and reduced data sparsity through joint training. Notable examples include USER (Yao et al., 2021), which models heterogeneous sequences of queries and

| Model | Architecture | Query-Aware Search | Query-Free Recommendation |
|---|---|---|---|
| HLLM (Chen et al., 2024) | Two Tier Decoder | × | ✓ (short-video feeds) |
| OneRec (Deng et al., 2025) | Encoder–Decoder MoE | × | ✓ (short-video feeds) |
| HSTU (Zhai et al., 2024) | Causal Encoder on ID-inputs | × | ✓ (Social Media) |
| MTGR (Han et al., 2025) | HSTU + dynamic masking | × | ✓ (food delivery) |
| Actions Speak (Zhai et al., 2024) | Trillion-scale fast-attention | × | ✓ (internet-scale feeds) |
| GenR-PO (Li et al., 2024) | Decoder-only | ✓ (e-commerce search) | × |
| GenRank (Huang et al., 2025) | Decoder-only | × | ✓ (recommendation) |
| UniSAR (Shi et al., 2024) | Unified decoder | ✓ | ✓ (but trade-off) |
| GenSAR (Shi et al., 2025) | Unified decoder | ✓ | ✓ (but trade-off) |
| *SynerGen* (ours) | Decoder-only | ✓ | ✓ |

Table 1: Comparison of representative generative recommender systems. Existing work typically supports only one mode (search or recommendation) or suffers from performance trade-offs when unifying them. *SynerGen* uniquely unifies retrieval and ranking across both modes without sacrificing effectiveness.

item interactions, and UnifiedSSR (Xie et al., 2023), which learns a shared representation of user behavior across both modalities. UniSAR (Shi et al., 2024) further refines this idea by explicitly modeling cross-modal transitions, using contrastive objectives and cross-attention to align and fuse behaviors from search and recommendation.

**Generative Search and Recommendation.** The emergence of generative large language models (LLMs) has catalyzed a new paradigm for unifying search and recommendation in e-commerce. Unlike traditional retrieval methods that rely on sparse or dense vector matching, generative LLMs can directly generate item identifiers or embeddings in response to user queries or profiles, enabling a seamless integration of both search and recommendation within a single framework. Recent advances demonstrate two primary approaches: text-based generative inputs (e.g., HLLM Chen et al. (2024)) and ID-based inputs (e.g., HSTU Zhai et al. (2024) and MTGR Han et al. (2025)). Both have shown that generative models can bridge the gap between search (semantic relevance) and recommendation (collaborative filtering).

Beyond these, a growing set of generative recommenders explore collapsing the retrieval–ranking pipeline altogether. OneRec (Deng et al., 2025) applies an encoder–decoder MoE architecture to short-video feeds, surpassing strong two-tower baselines. MTGR (Han et al., 2025) extends this idea to food delivery, using a hierarchical backbone with dynamic masking to preserve cross-feature interactions. Actions Speak Louder than Words (Zhai et al., 2024) scales generative recommendation to the trillion-parameter regime with specialized fast-attention kernels, achieving double-digit online lifts. In contrast, GenR-PO (Li et al., 2024) targets e-commerce search but does not support feed recommendation, while GenRank (Huang et al., 2025) focuses on recommendation without query signals. More recent attempts such as UniSAR (Shi et al., 2024) and GenSAR (Shi et al., 2025) aim at unified modeling but report trade-offs between query-aware and query-free modes.

**Time-Aware Sequence Modeling.** Incorporating temporal information into sequential recommendation allows models to better capture evolving user interests and short-term intent. The prevailing approach, used in seminal works such as SasRec Kang & McAuley (2018a) and Bert4Rec Sun et al. (2019), is to arrange event sequences chronologically. Beyond simple ordering, time can be injected directly through positional embeddings (absolute time) Makhneva et al. (2023) or through relative encodings in the attention mechanism. The latter has gained traction, with models such as HSTU Zhai et al. (2024) incorporating relative temporal bias into self-attention, enabling finer-grained modeling of temporal dependencies.

## 3 METHOD

In this section, we describe the proposed method in detail, beginning with the problem formulation and overall model overview.

## 3.1 PROBLEM FORMULATION AND MODEL OVERVIEW

We consider a user's historical interaction sequence $H = \{x_1, x_2, \ldots, x_T\}$, where each timestamped event $x_t$ is represented as a tuple of four heterogeneous token types:

$$x_t = (\text{ITEM}_t, \text{ACTION}_t, \text{TIME}_t, \text{QUERY}_t).$$

Here, ITEM denotes the product identity, ACTION specifies the interaction type (e.g., click, purchase, add-to-cart), TIME encodes the temporal context, and QUERY captures the associated search query or contextual signals.

Our objective is to jointly solve two core tasks of recommender systems within a single framework: (1) *Relevance estimation*, which retrieves potentially relevant items from the entire catalog, and (2) *Click probability estimation*, which ranks retrieved items by estimating the likelihood of user interaction.

*SynerGen* unifies these tasks in a generative paradigm by formulating them as conditional sequence generation problems. Specifically, retrieval corresponds to predicting the next relevant item given user history and intent, while ranking corresponds to estimating the probability of a specified action on a candidate item conditioned on history and query context.

## 3.2 INPUT REPRESENTATION AND MULTI-MODAL FUSION

**Embedding Construction.** Each interaction token $x_t$ is mapped into a continuous embedding space via a hybrid design that integrates semantic, collaborative, and contextual features:

$$\text{ITEM} = (e^{\text{item}_s}; e^{\text{item}_c}), \quad \text{ACTION} = e^{\text{action}}, \quad \text{QUERY} = e^{\text{query}}.$$

Semantic embeddings $e^{\text{item}_s}$ and $e^{\text{query}}$ are derived from a pretrained encoder LM (a 350M-parameter LLaMA-based model Touvron et al. (2023)), trained on item metadata, user queries, and query–item matching tasks with a joint masked language modeling and contrastive objective Zhang et al. (2025); BehnamGhader et al. (2024). This encoder produces $d$-dimensional semantic representations, where $e^{\text{item}_s}$ encodes product attributes (e.g., title, brand, color, material) and $e^{\text{query}}$ captures query semantics.

In parallel, collaborative embeddings $e^{\text{item}_c}$ (item IDs) and $e^{\text{action}}$ (interaction types) are randomly initialized and optimized during task-specific training. The combination of semantic signals (generalization) and ID-based signals (collaboration) ensures robustness to both frequent and long-tail items.

The final interaction representation $x_t$ is formed by concatenating all embeddings, which are projected to the model dimension via a fusion MLP, denoted as $\text{MLP}_{\text{fusion}}$.

## 3.3 DECODER BACKBONE ARCHITECTURE

*SynerGen* adopts a decoder-only Transformer with $L$ layers, hidden dimension $d$, and $H$ masked self-attention heads. The left-to-right causal model processes the sequence $x_t$ in an auto-regressive manner, which produces the next token by conditioning on past timesteps.

A key novelty in our architecture is the design of **task tokens**, which explicitly signal whether the model should perform context modeling, relevance estimation, or click probability estimation. These tokens allow the backbone to seamlessly switch between retrieval and ranking within the same sequence.

**Context Token.** The context token captures a user's behavioral history in a query-free manner, serving as the foundation for feed-style recommendation. By masking out query signals and retaining only item and action embeddings, the model learns a pure representation of evolving user intent:

$$e^{\text{context}} = \text{MLP}_{\text{fusion}}([0; e^{\langle\text{item}\rangle}; e^{\text{action}}]).$$

With the causal attetnion structure, this ensures that feed recommendation can be performed directly from behavioral dynamics, while also providing a clean state that query-aware retrieval and ranking can later condition on without leakage.

**Retrieval Token.**   The retrieval token is tailored for query-aware search. It takes the user's query and action type as input, but masks the item identity:

$$e^{\text{rel}} = \text{MLP}_{\text{fusion}}([e^{\text{query}}; e^{\langle\text{mask}\rangle}; e^{\text{action}}]).$$

Masking the item using a trainable embedding $e^{\langle\text{mask}\rangle}$ prevents trivial copying of the ground truth and forces the model to infer relevant items consistent with the query context. This mirrors real search scenarios, where the system must generate relevant candidates from a large catalog conditioned on query signals but without direct knowledge of the clicked item.

**Ranking Token.**   The ranking token supports both modes by calibrating probabilities of interaction with specific candidate items. Unlike retrieval, the candidate embedding is explicitly included but the query can be chosen to be presented or not:

$$e^{\text{rank}} = \text{MLP}_{\text{fusion}}([e^{\text{query}}; e^{\text{item}}; e^{\text{action}}]).$$

One can choose query to be present of nor for ranking task. In query-aware search, this design enables fine-grained click-probability estimation for items surfaced by retrieval; in query-free feed recommendation, the same ranking token evaluates each candidate's engagement likelihood purely from historical context. Explicitly supplying the candidate identity allows the model to perform calibrated probability estimation across both modes rather than mere sequence reconstruction. For the ranking token training, we include both positive and negative clicks: this balances the click vs. non-click label distribution and improves training efficiency, since mask-controlled attention lets positive and negative candidates share the same Transformer-encoded context while only the candidate representation changes.

**Unified Training and Inference.**   Retrieval and ranking are cast as conditional prediction problems: retrieval predicts relevant items given history, query, and action but without any item information, while ranking estimates interaction likelihood for given information on query, item, and history. During training, task tokens are inserted throughout sequences to provide dense supervision, exposing the backbone to both objectives. At inference, retrieval tokens score the entire catalog, while ranking tokens refine candidates with calibrated probabilities. This setup mirrors production pipelines, but within a single generative backbone, closing the gap between training and deployment.

### 3.4 TEMPORAL ATTENTION MECHANISM

**Task specific masking matrix**   To regulate information flow among heterogeneous tokens, we introduce a task-specific masking matrix $M$ that modifies the standard self-attention mechanism. In order to maintain the causality for each task, $M$ enforces three types of constraints. First, **temporal causality**: context tokens are only allowed to attend to strictly earlier events, ensuring chronological consistency in behavior modeling. Second, **session isolation**: retrieval and ranking tokens may only access historical context from previous request groups, preventing leakage of information from the current query session. Third, **cross-task alignment**: ranking tokens are additionally permitted to attend to retrieval tokens corresponding to the same interaction event, allowing them to reuse relevance signals when estimating click probabilities. The exact mathematical specification of our masking matrix $M$ is provided in Appendix.

**Modeling Time with RoPE**   User behavior unfolds over real time, with irregular intervals that simple sequential encodings cannot capture. To address this, we adopt Rotary Positional Embeddings (RoPE) Su et al. (2021) applied directly to Unix timestamps. Unlike discrete bucket encodings or absolute positional embeddings Vaswani et al. (2017), this approach encodes temporal gaps as rotations in the attention space. The design provides three benefits: (i) fine-grained modeling of absolute and relative time, (ii) shift invariance with respect to session start, and (iii) extrapolation to unseen time intervals. The full mathematical formulation of RoPE-based temporal attention, together with a worked example illustrating how absolute timestamps translate into relative rotations, is provided in Appendix.

## 3.5 MULTI-TASK TRAINING OBJECTIVES

We jointly optimize retrieval and ranking objectives within a single training loop, ensuring that both coarse-grained relevance estimation and fine-grained click probability estimation are aligned within the same backbone.

**Retrieval Loss.** The retrieval objective is formulated as an InfoNCE contrastive loss Oord et al. (2018), aligning retrieval token representations $h_i$ with their corresponding positive item embeddings $e_i^+$ while pushing them away from negatives:

$$\mathcal{L}_{\text{rel}} = \alpha \mathcal{L}_{\text{rel}}^{\text{easy}} + (1 - \alpha) \mathcal{L}_{\text{rel}}^{\text{hard}}.$$

Each component follows the standard InfoNCE form:

$$\mathcal{L}_{\text{rel}}^* = -\sum_{i=1}^{N_{\text{rel}}} \log \frac{\exp(\text{sim}(h_i, e_i^+)/\tau)}{\exp(\text{sim}(h_i, e_i^+)/\tau) + \sum_j \exp(\text{sim}(h_i, e_j^-)/\tau)},$$

where $\text{sim}(\cdot)$ denotes dot-product similarity and $\tau$ is a temperature parameter.

We adopt two types of negatives: (i) *in-batch random negatives*, which provide broad semantic contrast across items, and (ii) *impressed-but-not-clicked negatives*, which are items displayed to the user but ignored, representing challenging decision boundaries. This dual-negative strategy encourages both general discrimination (via easy negatives) and fine-grained preference modeling (via hard negatives).

**Ranking Loss.** For ranking, we construct positive–negative pairs using clicked (positive) and impressed-but-not-clicked (negative) observed in the same context. This design ensures that the model learns to distinguish between items that were actually presented to the user, rather than arbitrary negatives.

We employ a hybrid objective that combines pointwise and pairwise components. The *pointwise loss* $\mathcal{L}_{\text{point}}$ is a binary cross-entropy applied independently to each ranking token:

$$\mathcal{L}_{\text{point}} = -\sum_{i=1}^{N_{\text{rank}}} \Big[ y_i \log \sigma(s_i) + (1 - y_i) \log \big(1 - \sigma(s_i)\big) \Big], \quad \mathcal{L}_{\text{pair}} = -\sum_{(i,j) \in \mathcal{P}} \log \sigma(s_i - s_j),$$

where $s_i$ is the logit score predicted by the ranking head for token $i$ upon the decoder module, $y_i \in \{0, 1\}$ denotes the click label, and $\sigma(\cdot)$ is the sigmoid function. This objective encourages the model to output calibrated probabilities for click vs. non-click.

In addition, we apply a *pairwise loss* that directly enforces relative ordering between positive and negative candidates. For each positive–negative pair $(i, j)$, the loss takes the form as equation $\mathcal{L}_{\text{pair}}$, where $\mathcal{P}$ is the set of observed positive–negative pairs. This term encourages the score of the positive item to exceed that of the negative by a large margin.

The overall ranking loss combines the two objectives:

$$\mathcal{L}_{\text{rank}} = \mathcal{L}_{\text{point}} + \lambda \mathcal{L}_{\text{pair}},$$

where $\lambda$ controls the trade-off between probability calibration and relative ranking accuracy.

**Joint Optimization.** The final training objective combines retrieval and ranking losses:

$$\mathcal{L} = \mathcal{L}_{\text{rel}} + \mathcal{L}_{\text{rank}}.$$

This joint formulation tightly couples the two tasks: retrieval benefits from ranking's fine-grained supervision, while ranking leverages retrieval's catalog-level discrimination. Unlike conventional two-stage pipelines, our unified optimization ensures consistent representation learning and reduces system complexity.

## 4 EXPERIMENTS

In this section, we present a comprehensive evaluation of *SynerGen*. We begin by describing the experimental setup and evaluation metrics, followed by comparative results against state-of-the-art baselines. We then analyze various model components through ablation studies to understand their individual contributions.

Table 2: Performance of using **Book Review** dataset in recommendation task with all 686k Items as Candidates.

| Method | R@10 | R@50 | R@200 | N@10 | N@50 | N@200 |
|---|---|---|---|---|---|---|
| *SynerGen*-ID [†] | **7.33** | **17.49** | **30.39** | 3.91 | **6.11** | **8.09** |
| HLLM-1B-Scratch | 6.85 | 13.95 | 23.19 | **4.02** | 5.56 | 6.95 |
| HSTU-large [†] | 6.50 | 12.22 | 19.93 | 3.99 | 5.24 | 6.38 |
| SASRec [†] | 5.35 | 11.91 | 21.02 | 2.98 | 4.40 | 5.76 |
| *SynerGen** | **9.91** | **19.29** | **30.48** | 5.45 | 7.15 | 8.43 |
| Bert4Rec* | 4.58 | 12.54 | 24.08 | 1.94 | 3.66 | 5.39 |
| HLLM-1B* | 9.28 | 17.34 | 27.22 | **5.65** | **7.41** | **8.89** |

*Methods leverage pretrained semantic embeddings. [†] Methods trained only with collaborative embeddings (no semantic embeddings).

Table 3: Performance comparison on the **eBook Search Sessions** dataset across recommendation-only (left) and search-enhanced (right) settings using a candidate pool of 100 items (1 positive, 99 random negatives). Note that methods designed for only recommendation or only search are not applicable in the other setting and are thus omitted where appropriate.

| | Recommendation (no query) | | | | | Search (w/ synthetic query) | | | | |
|---|---|---|---|---|---|---|---|---|---|---|
| Method | R@1 | R@5 | R@10 | N@5 | N@10 | R@1 | R@5 | R@10 | N@5 | N@10 |
| *SynerGen* | **31.36** | **59.36** | **70.72** | **46.13** | **49.83** | **63.35** | **85.25** | 90.30 | **75.41** | **77.10** |
| UnifiedSSR | 20.13 | 51.96 | 67.07 | 36.62 | 41.51 | 36.63 | 77.44 | 88.12 | 58.47 | 61.96 |
| UniSAR | 30.10 | 58.74 | 70.20 | 45.13 | 48.85 | 53.43 | 81.90 | 89.77 | 68.75 | 71.32 |
| USER | 23.61 | 54.41 | 68.54 | 39.64 | 44.22 | 41.23 | 76.31 | 86.97 | 60.00 | 63.48 |
| SESRec | 27.26 | 56.23 | 68.64 | 42.45 | 46.48 | – | – | – | – | – |
| BERT4Rec | 24.81 | 53.11 | 66.58 | 39.54 | 43.90 | – | – | – | – | – |
| SASRec | 20.59 | 52.95 | 67.72 | 37.47 | 42.25 | – | – | – | – | – |
| TEM | – | – | – | – | – | 40.52 | 81.69 | **90.51** | 63.03 | 65.87 |
| CoPPS | – | – | – | – | – | 31.17 | 66.16 | 77.07 | 62.81 | 65.70 |

## 4.1 DATASETS AND EVALUATION PROTOCOLS

We highlight our evaluation for *SynerGen* on three representative datasets: **Book Review**, **eBook Search Sessions**, and **Session-US**. The statistics for each dataset are summarized in the appendix.

**Preprocessing.** For **Book Review** and **eBook Search Sessions**, we follow standard practice Chen et al. (2024): users and items with fewer than five interactions are removed, and a leave-one-out strategy is used to split sessions into training, validation, and test sets. Baseline results are reproduced under identical protocols for fair comparison.

**Book Review.** This dataset is used for sequential recommendation. Evaluation is performed over the full candidate set, with Recall@K and NDCG as metrics.

**eBook Search Sessions.** Constructed from large-scale interaction data, this dataset supports joint search and recommendation. Synthetic queries are generated following (Ai et al., 2017; 2019; Si et al., 2023; Shi et al., 2024). Each ground-truth item is paired with 99 randomly sampled negatives to form a candidate pool of 100. We report Recall@1,5,10 and NDCG@5,10, averaged over 10 runs with different seeds.

**Session-US.** A large-scale real-world e-commerce dataset derived from search logs (Wang et al., 2023). Interactions are chronologically ordered and segmented into weekly sub-sessions containing the final 100 events, with sessions of fewer than two actions discarded. Each record includes item IDs, timestamps, action types (clicks/purchases), and queries. The dataset exhibits realistic sparsity (average 60 interactions per session, long-tail item distribution). During training, the model processes roughly 75B relevance tokens and 150B ranking tokens per epoch.

Two benchmark tasks are defined: 1. *Contextualized Recommendation* — predict the next clicked or purchased item two days after training, without query information; evaluated by NDCG over 15 viewed items per impression. 2. *Contextualized Search* — predict the next click two days after training, with query information; evaluated by Recall over the full item set for retrieval and MRR

Table 4: Performance and ablation results across recommendation and search tasks on the Session-US dataset. For the *Full model*, all parameters are trained jointly; results are reported either by evaluating the **ranking head** or the **retrieval head** separately. All ablation experiments are evaluated via the retrieval head.

| | Retrieval (10M Pool) | | | Ranking tasks (Impressed pool) | |
| | Recommendation (query free) | | Search | Search | |
| Configuration | Click NDCG | Purchase NDCG | R@300 | MRR | R@1 |
|---|---|---|---|---|---|
| **Full model (trained with all parameters)** | | | | | |
| Full model (evaluated via Ranking head) | – | – | – | **58.20** | **33.84** |
| Full model (evaluated via Retrieval head) | **48.79** | **49.44** | 72.96 | 57.30 | 31.57 |
| **Ablations (evaluated via Retrieval head)** | | | | | |
| w/o Collaborative emb. | 48.21 | 47.44 | 61.59 | 56.68 | 30.83 |
| w/o Time-based RoPE | 48.56 | 48.12 | 70.56 | 57.05 | 31.25 |
| w/o Ranking head | 48.41 | 48.40 | **73.36** | 56.90 | 31.11 |
| w/o Target time info. | 48.66 | 49.29 | 71.36 | 57.19 | 31.54 |

over 7 viewed items for ranking. Both tasks employ $k$-nearest neighbor retrieval to assess the model's ability to capture user preferences under different contexts.

## 4.2 BASELINES

We compare *SynerGen* against three categories of baselines: (i) sequential recommendation models without search data, (ii) joint search–recommendation models, and (iii) personalized search models without recommendation data.

**Sequential recommendation.** On the Book Review dataset, we consider canonical sequence models SASRec (Kang & McAuley, 2018b) and BERT4Rec (Sun et al., 2019), along with more recent architectures. HLLM (Chen et al., 2024) introduces a hierarchical encoder for item content and user history, while HSTU (Zhai et al., 2024) represents a state-of-the-art sequential transducer for large-scale streaming data.

**Joint search and recommendation.** On the eBook Search Sessions dataset with synthetic queries, we include SESRec (Si et al., 2023), which leverages search interactions to improve recommendation; USER (Yao et al., 2021), which integrates queries and browsing into a single heterogeneous sequence; UnifiedSSR (Xie et al., 2023), which learns a shared representation of user history across both domains; and UniSAR (Shi et al., 2024), which models fine-grained transitions between search and recommendation via contrastive alignment and cross-attention fusion.

**Contextualized search.** We also compare with TEM (Bi et al., 2020), which incorporates user preferences and context for relevance estimation, and CoPPS (Dai et al., 2023), which aligns user intent with search results using contrastive learning.

## 4.3 IMPLEMENTATION AND TRAINING DETAILS

*SynerGen* is trained on 16 NVIDIA A100 GPUs with mixed precision. We use AdamW with learning rates of 0.001 (dense) and 0.003 (sparse), InfoNCE temperature $\tau = 0.085$, and loss weights $\alpha = 0.1$ (contrastive) and $\lambda = 1$ (pairwise). Unless otherwise noted, evaluation uses the retrieval head output. Different from trending generative recommender like Chen et al. (2024), our *SynerGen* is not jointly trained with the pretrained encoder LM.

Architectural and training hyperparameters are scaled to dataset size. For Book Review and eBook Search Sessions, we use a compact 100M-parameter decoder backbone; for Session-US, a larger 17B-parameter configuration is adopted, dominated by the collaborative embedding table. Additional dataset-specific settings (sequence length, negatives, RoPE granularity) for implementations are summarized in Appendix.

## 4.4 MAIN RESULTS

### 4.4.1 RECOMMENDATION

On the **Book Review** dataset with 686K candidate items (Table 2), *SynerGen* achieves the best or highly competitive performance across all metrics. Trained with only collaborative embedding, *Syn-*

*erGen*-ID attains the highest Recall, demonstrating the strength of the generative backbone. HLLM-1B-Scratch achieves slightly higher NDCG@10 (4.02), indicating better ranking of the top few items, but at substantially higher computational cost. Incorporating frozen semantic embeddings further improves *SynerGen*, confirming the benefit of semantic knowledge. Compared to HLLM-1B, which trains a 1B-parameter language model end-to-end, our 100M-parameter *SynerGen* attains higher Recall and nearly identical NDCG@10 (5.45 vs. 5.65), while being far more efficient. Freezing the semantic encoder reduces training cost without sacrificing retrieval or ranking quality.

### 4.4.2 JOINT SEARCH AND RECOMMENDATION

Results on the **eBook Search Sessions** dataset (Table 3) show that *SynerGen* consistently outperforms baselines in both recommendation-only and query-aware search settings. It achieves the highest Recall@1 and NDCG@10, demonstrating its ability to capture user intent across modalities. Against unified models such as UnifiedSSR, UniSAR, and USER, *SynerGen* delivers notable gains in early precision (Recall@1) and ranking quality (NDCG), particularly in the search scenario. Moreover, it surpasses specialized search systems such as TEM and CoPPS, underscoring the effectiveness of a single generative framework that jointly supports recommendation and search.

### 4.5 ABLATIONS

To assess the contribution of each component in *SynerGen*, we conduct an ablation study on the Session-US benchmark, covering both query-free recommendation and query-aware search. Results in Table 4 address the following questions:

**Q1: How do retrieval and ranking heads contribute across tasks?** Retrieval head outperforms on recommendation and recall, while ranking head is stronger on fine-grained ranking, showing their complementary roles in the full model.

**Q2: Are collaborative embeddings necessary?** Yes. Removing them and unfreezing the semantic encoder to compensate reduces search Recall@300 by over 11%. Collaborative signals from user–item interactions are therefore indispensable, while keeping the semantic encoder frozen preserves pretrained semantic knowledge as a regularizer.

**Q3: Does temporal modeling beyond sequential order matter?** Yes. Replacing time-aware RoPE with standard sequential encoding consistently hurts performance, showing that explicit timestamp modeling provides richer temporal context and better captures session dynamics.

**Q4: Is training ranking head essential?** Yes. Eliminating it slightly increases recall but sharply degrades ranking quality. The ranking head improves precision, regularizes retrieval representations, and independently achieves the best ranking results, validating our joint training design.

**Q5: Do target timestamps provide meaningful supervision?** Yes, albeit modestly. Removing them causes small but consistent drops in ranking metrics, indicating that target-aware temporal cues aid disambiguation, especially in sparse sessions.

These ablations confirm that collaborative embeddings, frozen semantic encoders, temporal modeling, and joint retrieval–ranking optimization are complementary, each contributing to balanced recall and precision across search and recommendation.

## 5 CONCLUSION

We presented *SynerGen*, a single generative backbone for unified personalized search and recommendation that jointly optimizes retrieval and ranking within a decoder-only Transformer architecture. Extensive experiments on public and large-scale industrial datasets demonstrate that *SynerGen* consistently outperforms strong sequential recommenders, unified search–recommendation models, and personalized search baselines, achieving state-of-the-art recall and ranking quality in both query-free and query-aware settings. Ablation studies confirm that each architectural component contributes to this performance, with collaborative embeddings, temporal modeling, and multi-head training proving particularly impactful. Future work includes extending the framework to multimodal inputs and exploring adaptive retrieval–ranking trade-offs for even more efficient deployment.

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
