# OpenReview forum: "SynerGen: Contextualized Generative Recommender for Unified Search and Recommendation"
_ICLR.cc/2026/Conference — ICLR 2026 Conference Withdrawn Submission_

### Official Review · Reviewer_WnLj · 2025-10-25

**Soundness:** 2
**Presentation:** 3
**Contribution:** 2
**Rating:** 4
**Confidence:** 4

**Summary:**

This work aims to unify the previously segregated retrieval and ranking processes in recommender systems by proposing SynerGen, which jointly learns to retrieve and rank and supports both query-free and query-aware recommendation. To this end the authors propose a Transformer-based architecture that uses different heads for retrieval and ranking. The former is trained using InfoNCE and the latter using a pairwise ranking loss. Furthermore, the authors use RoPE to incorporate time dependency in self-attention. The shown experiments demonstrate that SynerGen usually attains the best performance compared to several baselines.

**Strengths:**

- The proposed SynerGen is versatile and can perform retrieval and ranking in a single model, with support for query-aware and query-free settings.

- The results show strong performance for the proposed method.

**Weaknesses:**

**Novelty**

The authors claim that SynerGen is the first instance of a generative model that supports query-aware and query-free recommendation. This is not true, as there is prior work that supports query-aware and query-free recommendation, as mentioned in their related work section, e.g. UniSAR and GenSAR.
Moreover, the second contribution of unifying retrieval and ranking has also been explored in prior work that was not cited [1].
Finally, the application of RoPE as well as task aware embeddings are not novel, see [2][3]
Therefore, I recommend to carve out the contributions more precisely and convey them properly, for example the training paradigm may be a contribution combining ranking and retrieval losses.

[1]  Unifying Generative and Dense Retrieval for Sequential Recommendation, Yang et al., TMLR 2025

[2]  Recommender Systems with Generative Retrieval, Rajput et al., NeurIPS 2023

[3]  BERT: Pre-training of Deep Bidirectional Transformers for Language Understanding, Devlin et al., NAACL 2019

**Scalability**

In practice, recommender systems must deal with extremely large item and user sets while delivering decent performance efficiently. In line 245 the authors state that retrieval tokens score the entire catalog. This means that the cost of ranking scales linearly with the number of items, which gets expensive for large scale item sets. This is the reason why generative retrieval has been introduced, as it does not rely on scanning the entire item set. As scalability is imperative when it comes to recommendation systems, I strongly recommend to report inference times and memory requirements for SynerGen and competitors for the different datasets.

**Lack of comparison to generative baselines**

The authors claim that SynerGen is a generative model, but the experiments lack comparisons to generative baselines like TIGER [1], Mender [2],  or LiGER [3].

[1]  Recommender Systems with Generative Retrieval, Rajput et al., NeurIPS 2023

[2]  Preference Discerning with LLM-Enhanced Generative Retrieval, Paischer et al., TMLR 2025

[3]  Unifying Generative and Dense Retrieval for Sequential Recommendation, Yang et al., TMLR 2025


**Significance of results**

The authors did not include error bars which makes it difficult to judge the significance of the reported results. The gap to baselines is often marginal (Table 2 & Table 4), especially when looking at ablation studies. I highly recommend to also report error bars and ideally test for statistical significance.

**Presentation**

The authors present SynerGen as a generative model, however in line 245/246 the authors say that retrieval tokens store the entire catalog. This is confusing as if it was a truly generative approach, the recommendation set could simply be generated by the model as in [1], instead it is used in a discriminative manner.
Also there is a claim on SynerGen being a generative foundation model (line 61), but there is no supporting evidence that it actually follows the definition of a foundation model [2].
Finally, overclaiming in lines 442 and 446, TEM is better than SynerGen for R@10 in Table 3.

[1]  Recommender Systems with Generative Retrieval, Rajput et al., NeurIPS 2023

[2] On the opportunities and risks of foundation models, Bommasani et al., arXiv 2021

**Questions:**

See Weaknesses.

---

### Official Review · Reviewer_Ctbo · 2025-10-31

**Soundness:** 2
**Presentation:** 3
**Contribution:** 3
**Rating:** 4
**Confidence:** 2

**Summary:**

This paper proposes SynerGen, a decoder-only Transformer-based generative recommendation model aimed at unifying personalized search and query-free recommendation. It jointly optimizes the InfoNCE retrieval loss and hybrid pointwise-pairwise ranking loss, integrates semantic, collaborative, and temporal signals, and introduces a time-aware Rotary Positional Embedding (RoPE) to capture temporal dependencies. Experiments were conducted on the Book Review, eBook Search Sessions, and Session-US datasets. SynerGen outperformed existing baselines in metrics such as Recall@K (R@K) and NDCG, verifying the feasibility of a single generative backbone for industrial-scale information access.

**Strengths:**

1. It is the first to achieve dual unification of "query-aware search-query-free recommendation" and "retrieval-ranking" under a decoder-only architecture, solving the problem of single-task limitations or performance trade-offs in existing models. As shown in Table 1, SynerGen is the only model that supports dual modes without performance trade-offs

2. It combines frozen pre-trained semantic embeddings (capturing item/query semantics) with optimizable collaborative embeddings (capturing user-item interactions), balancing generalization ability and long-tail item handling. In experiments, SynerGen (with semantic embeddings) outperformed SynerGen-ID (using only collaborative embeddings)

**Weaknesses:**

1. The paper proposes "context/retrieval/ranking tokens" to enable task switching, but the ablation study does not compare the scheme of "no task tokens + unified encoding." It is impossible to rule out the possibility that "task unification can be achieved without dedicated tokens," and the irreplaceability of the token design lacks support

2. The experimental datasets are concentrated in "book reviews" and "e-commerce search." Although the paper mentions that existing models are applied in fields such as "short videos and food delivery," SynerGen has not verified its performance in these fields. It is impossible to confirm whether the model is suitable for recommendation scenarios dominated by non-text data

**Questions:**

1.Only the last interaction is used as label; intermediate signals in long sequences are ignored.

2.Ranking still relies on random negatives; harder adversarial negatives are not explored.

3.Absolute timestamp shifts may hurt RoPE; cross-domain generalization is not examined.

4.Task-token order is fixed; impact of different arrangements is not studied.

5.No latency–accuracy comparison with lightweight two-tower baselines.

6.Ablations remove components but do not analyze sensitivity of loss weights.

7.Performance under varying user-activity levels is not reported, leaving sparse-scene evidence incomplete.

---

### Official Review · Reviewer_tYZW · 2025-11-05

**Soundness:** 3
**Presentation:** 2
**Contribution:** 2
**Rating:** 2
**Confidence:** 5

**Summary:**

This paper proposes a generative recommender model, which can provide a single generative backbone for both personalized search and recommendation. Their decoder-only transformer can handle both retrieval and ranking tasks simultaneously. They demonstrated its performance using real-world datasets and compared it with other methods.

**Strengths:**

1.  The paper presents a novel generative recommender system (SynerGen) that unifies personalized search and recommendation. It addresses the challenge of integrating these two traditionally separate systems into a single framework, improving both efficiency and performance.

2. The model jointly optimizes retrieval and ranking tasks within the same architecture, making it computationally efficient while maintaining high performance.

3. The model shows impressive results on multiple benchmark datasets.

**Weaknesses:**

1. While SynerGen is shown to be more efficient than some large-scale models, it still requires significant computational resources, especially for training with larger datasets. This could limit its practical deployment for smaller companies or those with limited computational power.

2. Some sections of the paper, particularly the model architecture and ablation analysis, are highly technical and may be challenging for a broader audience. A simplified explanation of key innovations or a visual overview could make the paper more accessible to non-experts.

3. The paper compares with some baselines, but it is still insufficient; more GRMs should be added for comparison.

4. It seems that instructions for using LLM are missing.

**Questions:**

1. How does SynerGen perform in real-world industrial production environments compared to comprehensive benchmark tests?

2. What is the impact of freezing semantic embeddings on the model's flexibility?

3. How does SynerGen handle cold-start problems for new users or items?

---

### Official Review · Reviewer_NKkD · 2025-11-05

**Soundness:** 3
**Presentation:** 3
**Contribution:** 2
**Rating:** 4
**Confidence:** 4

**Summary:**

This paper introduces SynerGen, a decoder-only model that unifies retrieval and ranking tasks for both personalized search and recommendation. The authors address the limitation that existing generative recommender systems typically handle either query-aware search or query-free recommendation, but not both, by introducing task-specific tokens (retrieval, context, ranking). SynerGen employs joint optimization with InfoNCE loss for retrieval and a hybrid pointwise-pairwise loss for ranking, while incorporating time-aware RoPE for temporal dynamics.

**Strengths:**

S1: Practical relevance - addresses the important problem of unifying search and recommendation pipelines together with retrieval and ranking tasks with a clean, elegant solution.

S2: Comprehensive evaluation on academic datasets. The authors conduct tests on Book Review and eBook Search Sessions (synthetic dataset) with diverse baselines.

S3: Strong Empirical Results on Academic Benchmarks. SynerGen achieves improvements on Book Review and eBook Search Sessions datasets compared to existing baselines.

**Weaknesses:**

W1: Missing clearer justifications for why the proposed approach is superior. It's not clearly justified why the combined ranking or retrieval model outperforms a retrieval-only model only (with eg a pairwise auxiliary loss). For instance, what would be the retrieval NDCG in Table 4 obtained using a retrieval-loss only model?

W2: Missing contextualization wrt recent work, e.g., Zhang et al. Killing Two Birds with One Stone: Unifying Retrieval and Ranking with a Single Generative Recommendation Model. SIGIR’25.

W3. Experiments
a/ Missing baselines. While the paper conducts thorough experiments for academic datasets, there's no baseline for Session-US.
b/ eBook Search Sessions is synthetc.
c/ it's not clear if all models have been calibrated to have similar params/FLOPS budget (eg SynerGen uses 100M params, which AFAIK is a large model for the Amazon Books dataset). This should be clarified directly in the paper but isn't done consistently; eg HLLM-1B has parameter count, but others don't.

W4. The paper utilizes temporal RoPE and claims to have special mathematical formulation discussed the Appendix. However the submission doesn't have an Appendix.

**Questions:**

Can the authors provide a direct comparison between their unified model and a retrieval-only (or a ranking-only) baseline? This would help clarify the benefit of joint optimization over task-specific alternatives.

---

### Note · Authors · 2025-11-21

I have read and agree with the venue's withdrawal policy on behalf of myself and my co-authors.